# Risk Assessment of Pulegone in Foods Based on Benchmark Dose–Response Modeling

**DOI:** 10.3390/foods13182906

**Published:** 2024-09-13

**Authors:** Verena Voigt, Heike Franke, Dirk W. Lachenmeier

**Affiliations:** 1Postgraduate Study of Toxicology and Environmental Protection, Rudolf-Boehm-Institut für Pharmakologie und Toxikologie, Universität Leipzig, Härtelstrasse 16-18, 04107 Leipzig, Germany; wo92apag@studserv.uni-leipzig.de (V.V.); heike.franke@medizin.uni-leipzig.de (H.F.); 2Chemisches und Veterinäruntersuchungsamt (CVUA) Karlsruhe, Weissenburger Strasse 3, 76187 Karlsruhe, Germany

**Keywords:** pulegone, benchmark dose, acceptable daily intake, genotoxicity, cancer risk

## Abstract

This study presents a new risk assessment of pulegone, a substance classified as possibly carcinogenic to humans (Group 2B) by the WHO International Agency for Research on Cancer (IARC). The analysis used data from a two-year carcinogenicity studies in rats and mice conducted by the National Toxicology Program (NTP) in 2011. Because of the absence of a no-observed adverse effect level (NOAEL) in these studies, the benchmark dose (BMD) approach was employed as an alternative risk assessment method. The lowest BMD lower confidence level (BMDL) of 4.8 mg/kg b.w./day among the eight endpoints served as the point of departure for calculating an acceptable daily intake (ADI) of 48 μg/kg b.w./day. This new ADI is significantly lower than the previously established tolerable daily intake of 0.1 mg/kg b.w./day set in 1997. The analysis also considered various genotoxicity studies, which indicate that pulegone’s effects follow a nongenotoxic, thresholded mechanism. The estimated intake levels of pulegone in the European Union and USA were below the newly calculated ADI, suggesting a low health risk based on current consumption patterns.

## 1. Introduction

Pulegone is one of the main ingredients in European pennyroyal (*Mentha pulegium* L., see Figure 1) and American pennyroyal (*Hedeoma pulegiodes* L.), with concentrations of 85–97% (*w*/*v*) and 30% (*w*/*v*) in essential oils, respectively [1].

Pulegone occurs in lower concentrations in other mint species (*Mentha* spp.) and their essential oils. The aerial parts of this plant species have traditionally been used as a flavoring, spice, and tea. Pennyroyal oil is used in traditional medicine [1,2]. Other applications of pennyroyal oil and (pepper)mint oil include as flavoring in beverages, pastries, sweets, and ice cream, as well as use as a fragrance in detergents and cosmetics [1].

Pulegone ((5*R*)-5-methyl-2-propan-2-ylidenecyclohexan-1-one, CAS 89-82-7, see Figure 2) is a monoterpene ketone that naturally occurs in two enantiomers. In nature, the (*R*)-(+)-enantiomer is more abundant. A 1982 study investigating hepatotoxicity and pulmonary toxicity found that the (*S*)-(-)-enantiomer was significantly less toxic than its (*R*)-(+)-counterpart [3]. Subsequent research has primarily focused on the (*R*)-(+)-enantiomer. Consequently, this study exclusively focuses on the (*R*)-(+)-enantiomer of pulegone.

In a study by Anderson et al. [5], the ingestion of more than 10 mL of pennyroyal oil resulted in moderate to severe toxicity, whereas the ingestion of more than 15 mL of pennyroyal oil (approx. 250 mg/kg b.w. in a 60 kg woman) resulted in death. Typical symptoms were massive centrilobular necrosis, lung edema, internal bleeding, and weight loss [6,7,8]. Furthermore, 18 cases of hepatotoxicity were found in individuals who ingested 10 mL or more of pennyroyal oil. Lower intakes result in gastritis and mild toxic effects in the central nervous system [5].

Pulegone is mainly ingested through peppermint (*Mentha X piperita*) and cornmint (*Mentha arvensis* L.) [2]. The use of pennyroyal oil is no longer common in Europe because of its toxic effect [9]. Pulegone contents in varieties of *M. piperita* and *M. arvensis* oils range from 0.5% to 4.6%. Farley et al. found high concentrations of pulegone in young peppermint plants, but they decreased with increases in the ages of the plants and were metabolized to menthol and menthone [10].

Figure 3 presents an overview of the possible metabolites that can be formed from pulegone in rats and mice in vivo and may be responsible for adverse effects.

Maximum limits on the use of pulegone in foodstuffs are regulated by the European Union (EU). Furthermore, it is not permitted in Europe to add the substance itself to food [12,13]. There are no limits on the use of pulegone in medicinal products or preparations; the only quality-related criteria for plant preparations are specified in pharmaceutical monographs [4]. In the pharmaceutical monograph on peppermint oil, the highest recommended daily dose in the EU is 1.2 mL of peppermint oil [4]. It has been reported that doses of up to 2.3 mg/kg b.w./day can be expected for herbal medicinal products [4]. These reported levels of pulegone exceed the TDI (tolerable daily intake) of 0.1 mg/kg b.w./day in food established by the Committee of Experts on Flavoring Substances (CEFS), which is based on an NOAEL of 20 mg/kg b.w./day from a 28-day oral toxicity study in rats and an uncertainty factor of 200 [12].

The European Medicines Agency (EMA)’s public statement on the use of herbal medicinal products containing pulegone recommends an acceptable intake level of 0.75 mg/kg b.w./day based on data from a 2011 National Toxicology Program (NTP) study (three-month repeat-dose toxicity study). This value is higher than the older TDI (0.1 mg/kg b.w./day) in food [4]. In cosmetics, the concentration of pulegone should not exceed 1% [14]. Pulegone has not been approved as a synthetic flavoring agent in the USA [4,15].

The WHO International Agency for Research on Cancer (IARC) monographs program assessed the hazard of pulegone to cause cancer in 2016. In summary, the IARC considered pulegone as possibly carcinogenic to humans (Group 2B) [1]. This categorization is based on neoplastic findings in experimental animal studies (two–year mouse and rat gavage studies), which were conducted by the National Toxicology Program (NTP) [8]. As a possible carcinogenic mechanism, the authors of the NTP study believe that it acts as a genotoxic carcinogen in female rat bladders. However, an old study, conducted in the 1980s, did not demonstrate genotoxicity of pulegone [16]. The genotoxicity assays performed in the NTP study also showed no genotoxicity in two out of three assays, although the discrepancy in the third assay was not provided with a satisfactory explanation. Nevertheless, the authors of the NTP study consider it possible that pulegone is metabolically activated and forms reactive intermediates that can react with DNA (see Figure 3). According to the EMA, further detailed genotoxicity testing of pulegone or products containing pulegone is necessary for risk assessment [4]. Various approaches to determining the NOAEL for pulegone have been employed by JECFA and CEFS, and the values ranged between 440 μg/kg and 20 mg/kg [4]. Currently, the determination of the BDML (lower confidence limit of the benchmark doses) is preferred to the NOAEL for risk assessment. The BMDL can also be determined in the absence of an NOAEL, whereas an extra uncertainty factor must always be considered when determining an NOAEL via an LOAEL [17].

The EMA suggested the appropriateness of the BMD analysis of the two-year NTP study but that it did not provide a formal analysis. Therefore, the aim of this study was to fill this knowledge gap and calculate the BMDL from different endpoints of the NTP two-year study with mice and rats. Based on this BMDL as a POD (point of departure), the acceptable daily intake (ADI) for pulegone was calculated as part of a risk assessment.

## 2. Materials and Methods

The data for the determination of a BMDL in carcinogenic effects originate from the NTP-Report TR 563 “Toxicology and carcinogenesis studies of pulegone in F344/N rats and B6C3F1 mice” [8]. In this study, both rats and mice (50 males and 50 females) were administered pulegone by gavage for two weeks, for three months, and for two years, respectively. BMD modeling was performed using international guidelines [17,18]. The various calculations and models for determining the BMDL were carried out using the US EPA (Washington, DC, USA) BMDS software v. 3.3 [19]. The EPA recommendations regarding the criteria for determining a BMDL were taken into account. Further guidelines, such as the “Guidance of the use of the benchmark dose approach in risk assessment” of the European Food Safety Authority (EFSA) and “Modeling Dose response for risk assessment” of the International Programme On Chemical Safety (IPCS), were used [17,18]. The benchmark response was set to 10%, the risk type to “extra risk”, and a confidence interval of 95% was assumed. All dichotomous models were considered, and the models with the best *p*-value and lowest Akaike information criterion (AIC) were selected and recommended by the software. Finally, the ADI was determined using the calculated BMDL and the default uncertainty factor (UL) (ADI = BMDL/UL) [20].

## 3. Results

The most informative study on the toxicity of pulegone in animal bioassays was published in the context of the National Toxicological Program (NTP) in 2011 [8]. The NTP concluded that there is a clear indication of carcinogenic activity for pulegone in male and female B6C3F1 mice based on increased incidences of hepatocellular adenoma (for both sexes) and hepatoblastoma (for male mice).

The most important endpoints for risk assessment are the non-neoplastic and neoplastic changes. For significant changes resulting from the study, the BMDL values were determined using the BMDS software. Because of the IARC classification of pulegone as possibly carcinogenic to humans (Group 2B), only the most relevant neoplastic effects from the two-year study are presented in this article, and the BMDL and ADI were determined on this basis. Data on hepatocellular adenoma (including multiple) and hepatocellular adenoma, carcinoma, or hepatoblastoma (combined) in male mice show a significant change in the 75-mg/kg-dose group compared with the control group (see Table 1). For all three endpoints, the highest-dose group was excluded because the incidences in this dose group decreased and no plausible models could be proposed; thus, it was not possible to determine the BMDL with all of the data. According to technical guidance for the BMDS software, it is permitted to delete the highest-dose group if no plausible models can be determined, with the rationale being that data at the highest dose may be the least informative of responses in the lower dose region of interest [21]. The full modeling raw data are presented in Appendix A.

### 3.1. Multiple Hepatocellular Adenoma

In total, seven possible models were identified for the multiple hepatocellular adenoma endpoint using the BMDS software. The models multistage degrees 1 and 2, as well as quantal linear, were the recommended models; all three showed the same data for *p*-value, AIC, BMD, and BMDL. The alternative models were logistic and probit (see Table 2). For a better overview, a graphical visualization of only one model (multistage degree 2) is shown (Figure 4).

### 3.2. Hepatocellular Adenoma (Including Multiple)

In total, eight viable models were determined for the hepatocellular adenoma endpoint using the BMDS software. The log-logistic model was recommended because of its lowest AIC and highest *p*-value and was used as the basis for determining the BMDL (see Table 3). The graphical visualization of the selected model is shown in Figure 5.

### 3.3. Hepatocellular Adenoma or Carcinoma or Hepatoblastoma (Combined)

For the endpoint hepatocellular adenoma or carcinoma or hepatoblastoma (combined) in the two-year mouse study, the BMDS software generated a total of four viable models from the data. The two models multistage degree 1 and quantal linear were recommended by the software for determining a BMDL (see Table 4). The quantal linear model is shown in Figure 6.

## 4. Discussion

### 4.1. Calculated BMDL Values

The calculated BDML values are summarized in Table 5. Because of the classification of pulegone as a possible human carcinogen by IARC (Group 2B), only the neoplastic endpoints from the two-year study were considered in the calculation of a BMDL. Other non-neoplastic endpoints also showed significant changes in rats and mice, particularly hyaline glomerulopathy. This can be associated with various bladder tumors in rats, but it is very rare in humans and very unlikely to occur in humans [4]. Therefore, this endpoint was not considered to determine a BMDL and resulting ADI.

In the two-year study in rats, the authors of the NTP study derived an LOAEL of 18.75 mg/kg b.w. day based on various endpoints (e.g., nephropathy, hepatotoxicity, and incidences of urinary bladder papilloma). In the mouse studies, they derived an LOAEL of 37.5 mg/kg b.w./day based on various endpoints (including hepatocellular adenoma, hepatocellular carcinoma, and hepatoblastoma) [8].

In comparison, the calculated BMDLs were significantly lower than the LOAEL values. The lowest BMDL of 4.8 mg/kg b.w./day, calculated from the endpoint “multiple hepatocellular adenoma”, was used to calculate an ADI.

### 4.2. Risk Assessment of Pulegone

In a study on (*R*)-(+)-pulegone in mice, Gordon et al. [3] found that menthofuran is a major metabolite responsible for its toxic effect. The metabolism of pulegone and its toxic effect, based on the formation of menthofuran, appears to be well elucidated using various study approaches, but these studies were only based on animal experiments. Engel et al. [22], therefore, focused on human metabolism to clarify whether the metabolites found in rats and mice can be found in humans and whether new metabolites can be identified and characterized. There were differences compared with the animal studies. Menthofuran does not appear to play a role in humans, at least at low doses, and 10-hydroxypulegone is preferentially formed, which is converted into the reactive 10-pulegone aldehyde. The most probable metabolism of pulegone occurs via the following three pathways: (i) hydroxylation with subsequent glucuronidation; (ii) reduction to menthone/isomenthone and subsequent hydroxylation or glucuronidation; and (iii) formation of mercapturic acids with subsequent hydroxylation [11] (Figure 3).

The mode of action is important in the risk assessment of carcinogenic substances. A carcinogenic substance can have a genotoxic or nongenotoxic mechanism. Genotoxicity refers to the interaction or reaction of a foreign substances with genetic material, triggering mutations. These mutations can occur at the gene level (point mutations) and at the chromosome level (aberrations) or genome level. Genotoxicity can also be divided into direct and non-direct genotoxicity. A direct genotoxic substance reacts directly with DNA (with adduct formation) without prior activation, whereas indirect genotoxic substances typically only trigger effects after prior metabolic activation. Nongenotoxic substances react with non-DNA target structures (such as topoisomerases or spindle proteins) or affect gene expression, growth reactions, or cell survival. Furthermore, it is important whether or not a so-called threshold value can be defined for a carcinogenic substance. The threshold value indicates the concentration of the dose below which no negative effects are expected. For carcinogenic substances without a threshold value, there is therefore no minimum concentration, i.e., the intake of a carcinogenic substance can lead to a change in the genetic material and, thus, to the initiation of cancer cells, regardless of the concentration or dose. Typically, indirect genotoxic substances are subject to a threshold value, whereas direct genotoxic substances do not have a threshold value [23,24,25,26].

Jenkins et al. [27] suggested that a threshold value could exist for direct genotoxic substances at low concentrations. DNA repair exists as one of several protective mechanisms against genotoxic substances in the human body. In their study, Jenkins et al. assumed that DNA repair takes place at low concentrations and can thus prevent changes to the genetic material. However, at higher concentrations of a genotoxic substance, this mechanism may become saturated and, therefore, no longer sufficient. The authors of the article showed that for two well-known alkylating substances methyl methanesulfonate (MMS) and ethyl methanesulfonate (EMS), there could also be a threshold value for direct genotoxic substances at low doses. In their opinion, such studies should be conducted for other substances in the future [27]. The presence or absence of a threshold value is important for the risk assessment of toxic compounds. If a corresponding threshold value is present, an ADI (acceptable daily intake) can be determined for the substance. The ADI is the concentration of a substance that can be ingested daily for a lifetime without having adverse health effects [28].

To determine the mutagenicity or genotoxicity of pulegone, Anderson and Jensen performed the AMES test in 1984. None of the *Salmonella* strains tested (TA1537, TA98, TA1535, and TA100) were found to have mutagenic properties or activities [16]. A total of three independent genotoxicity tests were carried out in the 2011 NTP study. The first two tests were negative with and without metabolic activation by the S9 mix. The second test was performed using the same lot of pulegone as in the two-year study. The third test, also performed with the same lot of pulegone as in the two-year study, showed positive results for the bacterial strains *Salmonella typhimurium* TA98 and *E. coli* WP2 uvrA/pKU101 with metabolic activation (10% S9 Mix). A possible explanation for the positive result may be contamination with menthofuran, a known pulegone metabolite. Menthofuran can be metabolized to a γ-ketoenal or an epoxyfuran ring in the presence of CYP enzymes. These substances could have led to positive results [29]. The degradation of glutathione caused by pulegone may also have led to a reduced detoxification of the active metabolites, leading to a positive result [29]. A more recent study, conducted in 2020, examined both pulegone and peppermint oil for possible mutagenicity [30]. In this study, too, no mutagenicity of pulegone or peppermint oil containing pulegone was detected in any of the *Salmonella* strains tested, both with and without metabolic activation.

In addition to the Ames tests described, used to determine possible genotoxicity, various tests were carried out with peppermint oil and not with pulegone itself. These tests were the Ames test, an in vitro mouse lymphoma test, and the in vitro rat bone marrow micronucleus test. The peppermint oil used contained 0.9% pulegone [4]. All three tests were negative; thus, peppermint oil showed no genotoxic potential. Further tests (i.e., COMET assay in lymphocytes and micronucleus assay in peripheral erythrocytes) with mint oil and peperina oil (essential oil of *Minthostachys verticillata*) also showed no genotoxic potential [31]. Considering all of the tests performed, the EMA concluded that both pulegone and menthofuran have no genotoxic potential [4].

Although the authors of the NTP study believe pulegone exhibits a genotoxic mechanism in female rat bladders, the IARC assumes a nongenotoxic mode of action based on the available data. Furthermore, the EMA indicated that tumors occurring in test animals are not relevant for the risk assessment of carcinogenicity in humans and that the data can be used for the determination of an NOAEL or BMDL [4].

The authors of this study are of the same opinion as the IARC, because of the previous negative studies and the negative study on the genotoxicity of pulegone published in 2020. Therefore, a nongenotoxic mechanism is assumed, which explains why the ADI can be determined for risk assessment of pulegone in humans.

### 4.3. Calculation of ADI

To determine the ADI of a substance, the determined BMDL value is divided by a so-called uncertainty factor. Typically, an uncertainty factor of 100 is assumed, which consists of the following: 10 (interspecies differences) × 10 (intraspecies differences). If an NOAEL (no-observed adverse effect level) cannot be derived from the study, but only an LOAEL (lowest observed adverse effect level), an uncertainty factor of between 3 and 10 is also taken into account for extrapolation to an NOAEL [18,32], such that an uncertainty factor of between 300 and 1000 would be assumed for the LOAEL of 37.5 mg/kg b.w./day available from the NTP study. This would result in ADI values between 125.0 μg/kg b.w./day and 37.5 μg/kg b.w./day, respectively.

The EMA calculated an NOAEL for pulegone in herbal medicinal products based on the NTP’s three-month repeat dose toxicity study in rats due to the following aspects: (1) the rat toxicity study that was used by CEFS to calculate a TDI of 0.1 mg/kg b.w./day is not suitable for determining an NOAEL due to its short duration (28 days) and the lack of detailed information; (2) carcinogenicity studies should not be used when considering non-neoplastic findings, so the two-year study is not considered; and (3) a three-month study in rats with peperina oil, published by Escobar et al. [31], is disadvantageous because of the administration via diet (uncertain exposure), use of a mixture, and lack of a complete histopathological evaluation of the NTP two-month study [31]. The NOAEL was at 37.5 mg/kg b.w./day based on the liver and kidney toxicities of the NTP’s three-month study. To calculate the acceptable intake level, an uncertainty factor of 50 was assumed instead of the usual 100. The authors of the statement justified this reduced uncertainty factor by the fact that the reaction pattern regarding acute liver toxicity, potential formation of active metabolites, and glutathione degradation was comparable between rats and humans. This resulted in an acceptable intake level of 0.75 mg/kg b.w./day [4].

Based on this rationale by EMA, a reduction in the uncertainty factor could also be considered in this study. In this case, however, the authors do not consider it appropriate for the following reasons: (1) the data from the NTP’s two-year study were used instead of the NTP’s three-month study; (2) the classification of pulegone as possibly carcinogenic to humans (group 2B), only neoplastic findings were considered; (3) the EMA concluded that tumors developing at high pulegone concentrations are not relevant in humans and thus comparability among the species is no longer provided. Since a BMDL instead of an LOAEL was used in this article employing data from the NTP study, the extra uncertainty factor for extrapolation from the LOAEL to the NOAEL can be omitted, so that an uncertainty factor of 100 was used. Thus, an ADI of 48 μg/kg b.w./day can be established for pulegone.

The calculated ADI of 48 μg/kg b.w./day is higher than the exposure in Europe and the USA. For example, according to the Joint FAO/WHO Expert Committee on Food Additives (JECFA), the estimated exposure of pulegone via food and cosmetics is 2 μg/day or 0.04 μg/kg b.w./day in Europe and 12 μg/day or 0.03 μg/kg b.w./day in the USA [4]. In France, intake estimates within a one-year household budget survey for pulegone in chewing gum, confectionery, and alcoholic beverages were 0.05 mg/day or 0.83 μg/kg b.w./day (recalculated for a 60 kg bodyweight) [7,12]. More specific data from the UK within the National Diet and Nutritional Survey of British Adults show a mean estimated intake of 0.03 mg/kg b.w./day for food and drink and 0.1 mg/kg b.w./day for the 97.5th percentile. It should be noted, however, that the estimates refer to the total intakes of pulegone and menthofuran [12], as the Committee is of the opinion that menthofuran is an important metabolite of pulegone and plays a role in toxicity, and that it is therefore necessary to consider both substances together. As this article only calculated the BMDL and ADI for pulegone using data from the NTP study, the exposure data described from the UK were not informative for the assessment.

### 4.4. Relevance of Neoplastic Lesion Findings in Humans

For the ADI for pulegone in food calculated in this article, the neoplastic endpoints hepatocellular adenoma and carcinoma were used from the 2011 NTP study. These types are often considered rodent-specific tumors. Another rather unusual adverse effect within the 2011 NTP study was hyaline glomerulopathy, which is very rare in humans. The EMA is therefore of the opinion that tumors occurring in animals (rats and mice) have no relevance to humans [4].

The EMA considers chronic regenerative cell proliferation a possible mode of action for the types of cancer found in rodents. Da Rocha et al. [33] presented the following key events for the mode of action of pulegone in female rat bladders in a publication: (1) chronic exposure to high concentrations of pulegone; (2) metabolism, excretion, and concentration of pulegone and cytotoxic metabolites, especially piperitenone, in urine; (3) urothelial cytotoxicity; (4) sustained regenerative urothelial cell proliferation; and (5) development of urothelial tumors [33]. Even if comparable metabolites occur in human urine [22], chronic exposure to high pulegone concentrations (75 mg/kg and higher) is not plausible, as such concentrations lead to nasal irritation and damage to the nose. Therefore, the mode of action for bladder tumors does not appear relevant to humans [33]. Nevertheless, data on the rodent tumors found can be used for the risk assessment of pulegone in humans to determine the NOAEL or BMDL, as stressed by EMA [4].

## 5. Conclusions

The POD determined using the BMD approach for pulegone was 4.8 mg/kg b.w./day. This value was lower than the POD of 37.5 mg/kg b.w./day (NTP study) or 20 mg/kg b.w./day (CEFS), previously determined using the LOAEL or NOAEL. From the newly calculated POD of 4.8 mg/kg b.w./day, an ADI of 48 μg/kg b.w./day was calculated, which is significantly higher than the estimated intake levels of pulegone from food and beverages in the EU and the USA; thus, none or only a very low health risk can be assumed. A limitation of this study is the uncertainty introduced by interspecies differences in pulegone metabolism between rodents and humans, which could affect the extrapolation of rodent data to human risk, highlighting the need for further research into human-specific pulegone metabolism to refine future risk assessments. Although this study focused exclusively on pulegone, some regulatory bodies consider the combined risk of pulegone and its metabolite menthofuran. A comprehensive risk assessment including both compounds, though beyond the scope of this current work, represents an important avenue for future research and could provide a more complete picture of the potential health risks associated with exposure to these substances.

## Figures and Tables

**Figure 1 foods-13-02906-f001:**
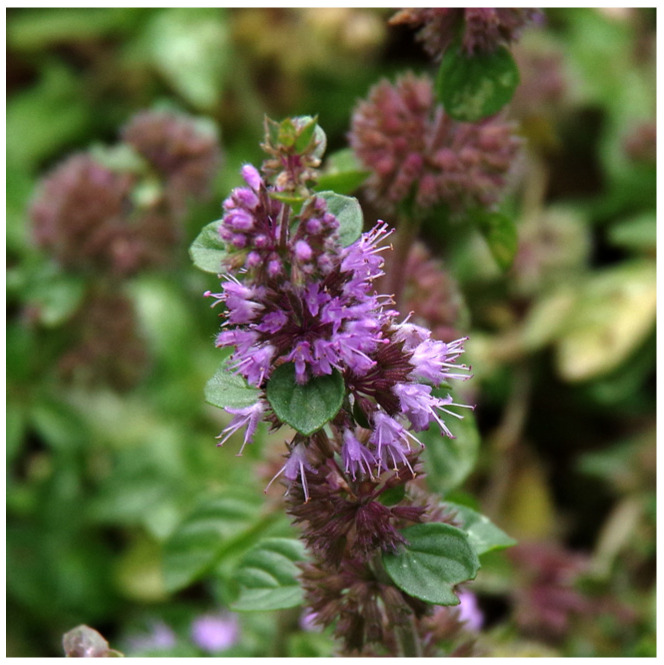
*Mentha pulegium* L.

**Figure 2 foods-13-02906-f002:**
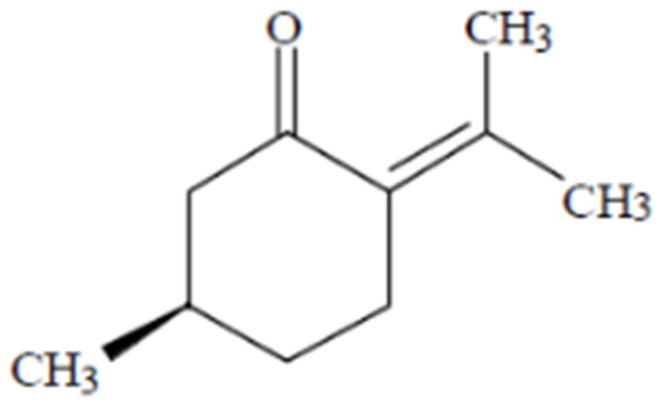
Chemical structure of (*R*)-(+)-pulegone [4].

**Figure 3 foods-13-02906-f003:**
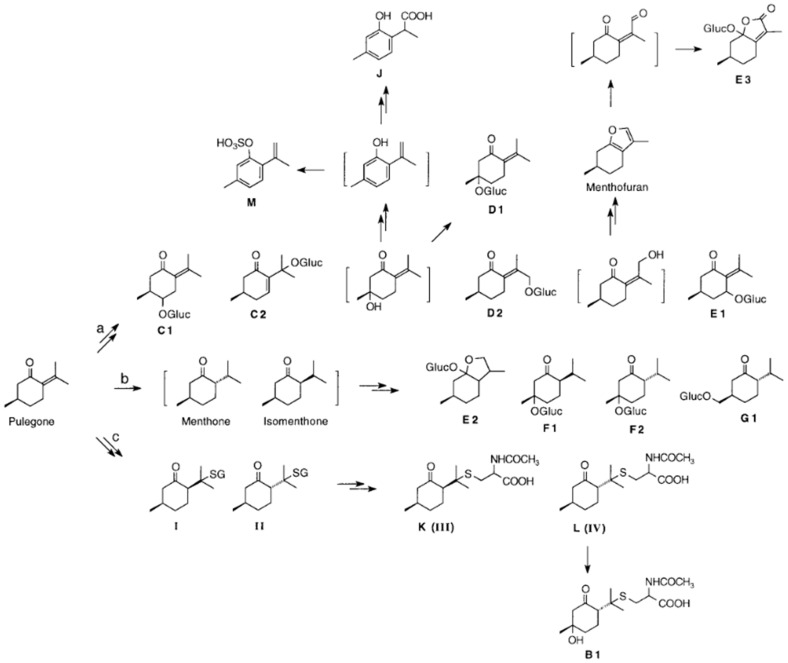
Metabolic pathways proposed for the biotransformation of pulegone in mice and rats [11]. Pathways: (a) hydroxylation followed by glucuronidation (metabolites C1, D1, D2, and E1) or further metabolism to C2, E3, J, or M; (b) reduction to (iso)menthone, followed by hydroxylation/glucuronidation (metabolites E2, F1, F2, and G1); (c) formation of glutathione conjugates (I or II), followed by conversion to mercapturic acids (metabolites K(III) or L(IV)) [11].

**Figure 4 foods-13-02906-f004:**
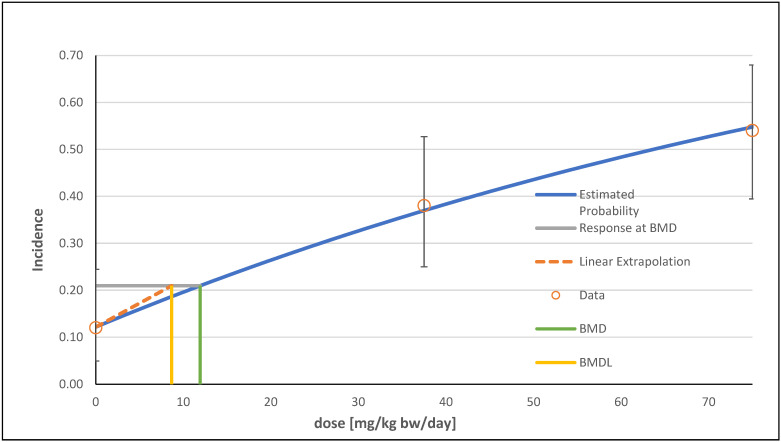
Frequentist multistage degree 2 model with a BMR of 10% extra risk for the BMD and a 0.95 lower confidence limit for the BMDL for the endpoint multiple hepatocellular adenoma in male mice.

**Figure 5 foods-13-02906-f005:**
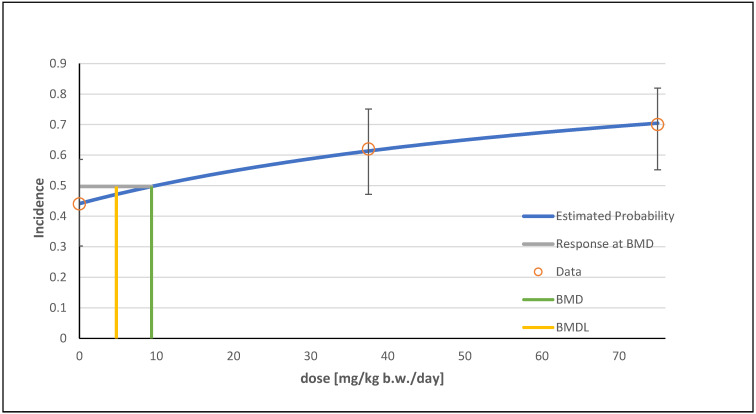
Frequentist log-logistic model with a BMR of 10% extra risk for the BMD and a 0.95 lower confidence limit for the BMDL for the endpoint hepatocellular adenoma (including multiple).

**Figure 6 foods-13-02906-f006:**
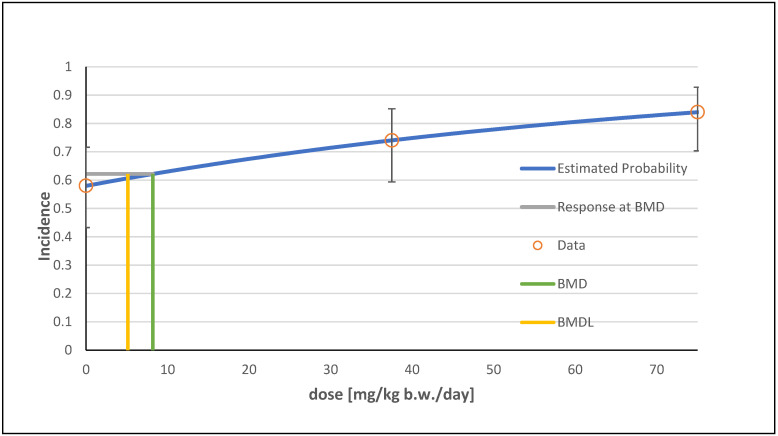
Frequentist quantal linear model with a BMR of 10% extra risk for the BMD and a 0.95 lower confidence limit for the BMDL for the endpoint hepatocellular adenoma or carcinoma or hepatoblastoma (combined).

**Table 1 foods-13-02906-t001:** Results of different neoplastic lesions in mice and rats from a 2-year gavage study performed by the National Toxicology Program (NTP).

Male Mice	Vehicle Control	37.5 mg/kg b.w./day	75 mg/kg b.w./day	150 mg/kg b.w./day
Hepatocellular adenoma (includes multiple)	22/50 (44%)	31/50 (62%)	35/50 * (70%)	28/50 (56%)
	*p* = 0.175 ^1^	*p* = 0.058 ^1^	*p* = 0.008 **^1^	*p* = 0.150 ^1^
Hepatocellular adenoma, carcinoma, or hepatoblastoma (combined)	29/50 (58%)	37/50 (74%)	42/50 * (84%)	36/50 (72%)
	*p* = 0.038 **^1^	*p* = 0.064 ^1^	*p* = 0.004 **^1^	*p* = 0.051 ^1^
Multiple hepatocellular adenoma	6/50 (12%) *p* ≤ 0.01	19/50 * (38%) *p* ≤ 0.01	27/50 * (54%) *p* ≤ 0.01	18/50 * (36%) *p* ≤ 0.01

* Statistically significantly different from the vehicle control group (*p* ≤ 0.05); for more details, see NTP report 563 [8]. ** *p* ≤ 0.01. ^1^ Poly 3 test—beneath the vehicle group: *p*-value associated with the trend test; beneath the dose groups: results of the comparisons between the vehicle group and dose group.

**Table 2 foods-13-02906-t002:** Benchmark dose modeling results for the endpoint multiple hepatocellular adenoma.

Model	*p*-Value	AIC	BMD	BMDL	Comment
Multistage degree 1, multistage degree 2, and quantal linear	0.849	176.13	11.9	8.7	Recommended models by the BDMS Software (lowest AIC)
Logistic	0.272	177.29	20.7	16.9	Viable model—alternate
Probit	0.315	177.10	19.6	16.1	Viable model—alternate

AIC = Akaike information criterion; BMD = benchmark dose; BMDL = lower confidence limit of the benchmark dose; BMDS = Benchmark Dose Software v 3.3.

**Table 3 foods-13-02906-t003:** Benchmark dose modeling results for the endpoint hepatocellular adenoma (including multiple).

Model	*p*-Value	AIC	BMD	BMDL	Comment
Log-logistic	0.905	200.1	9.4	4.8	Recommended model by the BDMS software (lowest AIC)
Gamma, multistage degree 1 and 2, Weibull, and quantal linear	0.728	200.2	12.2	7.4	Viable models—alternate
Logistic	0.603	200.4	14.9	10.3	Viable model—alternate
Probit	0.592	200.4	15.1	10.5	Viable model—alternate

AIC = Akaike information criterion; BMD = benchmark dose; BMDL = lower confidence limit of the benchmark dose; BMDS = Benchmark Dose Software v. 3.3.

**Table 4 foods-13-02906-t004:** Benchmark dose modeling results for the endpoint hepatocellular adenoma or carcinoma or hepatoblastoma (combined).

Model	*p*-Value	AIC	BMD	BMDL	Comment
Multistage degree 1 and quantal linear	0.992	173.30	8.2	5.1	Recommended model by the BDMS software (lowest AIC)
Logistic	0.896	173.32	9.7	6.7	Viable model—alternate
Probit	0.848	173.34	10.1	7.1	Viable model—alternate

AIC = Akaike information criterion; BMD = benchmark dose; BMDL = lower confidence limit of the benchmark dose; BMDS = Benchmark Dose software v. 3.3.

**Table 5 foods-13-02906-t005:** Overview of the most relevant endpoints from the 2-year gavage study performed by the NTP to calculate a BMDL.

Endpoint	Species	Sex	NOAEL or LOAEL (mg/kg b.w./day)	BDML (mg/kg b.w./day)
Multiple hepatocellular adenoma	Mice	Male Female	LOAEL 37.5 ^1^	8.7 ^2^
Hepatocellular adenoma (including multiple)	Mice	Male Female	LOAEL 37.5 ^1^	4.8 ^2,3^
Hepatocellular adenoma or carcinoma or hepatoblastoma (combined)	Mice	Male	LOAEL 37.5 ^1^	8.2 ^2^
Hyaline glomerulopathy (kidney)	Mice	Male Female	LOAEL 37.5 (male) ^4^ LOAEL 75.0 (female) ^4^	6.9 (male) 29.6 (female)
Hyaline glomerulopathy (kidney)	Rats	Male Female	LOAEL 18.75 ^1^	28.4 (male) 25.9 (female)

^1^ LOAEL values according to the NTP [8]. ^2^ Calculated BMDL values using an approach that includes skipping the high-dose group (150 mg/kg b.w./day). ^3^ BMDL proposed by the authors of this article for use as a point of departure (POD). ^4^ LOAEL for these endpoints not available from the NTP [8]. Therefore, the values were estimated by the authors of this article based on raw data published by the NTP.

## Data Availability

The original contributions presented in the study are included in the article/Appendix A, further inquiries can be directed to the corresponding author.

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
