# Peer review of "Risk Assessment of Pulegone in Foods Based on Benchmark Dose–Response Modeling"

_foods, 2024, doi:10.3390/foods13182906_

Round 1

Reviewer 1 Report

Comments and Suggestions for Authors

Overall the manuscript is good and presents an interesting study based on data from other studies, however, some recommendations are made: 

The abstract is very theoretical, it should briefly explain what was done in the article and what was found. The introduction is too long. 

The sentence in lines 100-101 is too short, it should be linked to the others.

Improve the wording, there are loose sentences and paragraphs that should be joined with the previous ones. 

The discussion lacks more citations. 

Of all the work that was done and the models that were generated, a validation could be done? because they are based on specific data. 

Author Response

Overall the manuscript is good and presents an interesting study based on data from other studies, however, some recommendations are made: 

The abstract is very theoretical, it should briefly explain what was done in the article and what was found.

RESPONSE: The abstract was completely revised considering the reviewers suggestion. It was shortened and the logical flow improved.

The introduction is too long. 

RESPONSE: The introduction was shortened by removing the exposure aspect and some further minor points.

The sentence in lines 100-101 is too short, it should be linked to the others.

RESPONSE: The sentence was linked to the previous paragraph as requested.

Improve the wording, there are loose sentences and paragraphs that should be joined with the previous ones. 

RESPONSE: A thorough English language check was conducted. The short paragraphs were linked with the previous ones as requested.

The discussion lacks more citations. 

RESPONSE: We appreciate the reviewer's attention to the citation practices in our discussion section. Upon careful consideration, we believe that the current level of citation adequately supports our arguments and findings. The discussion incorporates 17 distinct references, covering a wide range of relevant topics from metabolism studies to risk assessment methodologies. We have made efforts to include recent research, such as a 2020 study on mutagenicity, alongside established sources from respected organizations like the NTP, IARC, and EMA. Our aim was to strike a balance between presenting our own findings and contextualizing them within the broader scientific literature. However, we are open to suggestions for specific areas where additional citations might strengthen the discussion further. However, searches did not reveal new pertinent references on the topic.

Of all the work that was done and the models that were generated, a validation could be done? because they are based on specific data. 

RESPONSE: We appreciate the reviewer's insightful comment regarding validation of our work. It's important to clarify that our analysis is primarily based on the National Toxicology Program (NTP) 2-year carcinogenicity studies, which currently represent the most comprehensive long-term data available for pulegone toxicity. While similar long-term studies for direct validation are lacking, it's noteworthy that our results are in reasonable agreement with previous findings based on short-term studies, providing a degree of corroboration.

Reviewer 2 Report

Comments and Suggestions for Authors

The manuscript by Voigt et al. presents a complete risk assessment of pulegone in foods using benchmark dose (BMD) modeling. The study addresses a public health concern, given the potential carcinogenicity of pulegone, and provides updated acceptable daily intake (ADI) levels based on recent toxicological data. The paper is well-structured, and the methodology is appropriate, providing valuable insights into the risk assessment of pulegone. Interestingly, the utilization of BMD modeling instead of traditional NOAEL approaches is a significant strength, as it allows for a more accurate and data-driven risk assessment. Also, the manuscript gives a good analysis of important toxicological studies, including neoplastic and non-neoplastic endpoints.

Also, I would like to give the authors some suggestions for improvement:

1.    While the authors acknowledge that the tumors observed in rodent studies may not be directly relevant to humans, further discussion on the potential human relevance of these findings would be beneficial.

2.    The rationale for applying an uncertainty factor of 100 should be elaborated. Given that the EMA used a reduced uncertainty factor of 50 in a related context, the manuscript should provide a stronger justification for the chosen factor or discuss the potential implications of using a different factor.

3.    The study primarily focuses on pulegone but given the potential role of menthofuran as a toxic metabolite, the manuscript should consider the combined risk of pulegone and menthofuran exposure, as some regulatory bodies do.

Furthermore, I would like to present some questions for the authors:

Can the authors provide more details on the specific criteria used to exclude the highest dose group in the BMD modeling?

Have the authors considered the implications of interspecies differences in metabolism when interpreting the rodent data for human risk assessment? Specifically, how might differences in pulegone metabolism between humans and rodents affect the study's conclusions?

Given the IARC's classification of pulegone as possibly carcinogenic to humans (Group 2B), how do the authors propose that regulatory agencies balance the findings of this study with other potential carcinogens in the food supply?

Author Response

The manuscript by Voigt et al. presents a complete risk assessment of pulegone in foods using benchmark dose (BMD) modeling. The study addresses a public health concern, given the potential carcinogenicity of pulegone, and provides updated acceptable daily intake (ADI) levels based on recent toxicological data. The paper is well-structured, and the methodology is appropriate, providing valuable insights into the risk assessment of pulegone. Interestingly, the utilization of BMD modeling instead of traditional NOAEL approaches is a significant strength, as it allows for a more accurate and data-driven risk assessment. Also, the manuscript gives a good analysis of important toxicological studies, including neoplastic and non-neoplastic endpoints.

Also, I would like to give the authors some suggestions for improvement:

  1. While the authors acknowledge that the tumors observed in rodent studies may not be directly relevant to humans, further discussion on the potential human relevance of these findings would be beneficial.

RESPONSE: Thank you for bringing this to our attention. We appreciate the opportunity to address this comment and clarify our discussion on the human relevance of the rodent study findings. We understand the reviewer's desire for further elaboration on this important topic. In response to the reviewer's comment, we would like to emphasize that we have indeed dedicated a full section (4.4) to discussing the relevance of neoplastic lesion findings for humans. This section provides a comprehensive analysis of the topic, including identification of the tumor types (hepatocellular adenoma and carcinoma) used for ADI calculation, noting their consideration as often rodent-specific; discussion of hyaline glomerulopathy, an unusual adverse effect rarely seen in humans; presentation of the EMA's opinion that these tumors in rodents may not be relevant for humans; detailed explanation of the proposed mode of action for pulegone in female rat bladders, as described by Da Rocha et al. [33]; analysis of why this mode of action is unlikely to be relevant in humans, due to the implausibility of chronic exposure to high pulegone concentrations in humans; and acknowledgment that despite potential lack of direct relevance, these data can still be valuable for risk assessment in humans (according to EMA). We believe this section provides a thorough discussion of the potential human relevance of the rodent study findings.

2.    The rationale for applying an uncertainty factor of 100 should be elaborated. Given that the EMA used a reduced uncertainty factor of 50 in a related context, the manuscript should provide a stronger justification for the chosen factor or discuss the potential implications of using a different factor.

RESPONSE: Thank you for the opportunity to address this comment. We believe that our manuscript already provides a comprehensive rationale for applying an uncertainty factor of 100 (section 4.3), but we appreciate the chance to clarify and emphasize our justification. In response to the reviewer's comment, we would like to highlight the following points: We have indeed considered the EMA's approach of using a reduced uncertainty factor of 50 and explicitly discussed it in our manuscript. We provide clear reasons for why we do not consider the reduced factor appropriate in our case: We used data from the NTP 2-year study rather than the 3-month study used by the EMA; we focused only on neoplastic findings due to pulegone's classification as possibly carcinogenic to humans (Group 2B); and the EMA's conclusion that tumors developing at high pulegone concentrations are not relevant in humans undermines the species comparability that justified their reduced factor. We explain that our use of a BMDL instead of a LOAEL eliminates the need for an extra uncertainty factor for LOAEL to NOAEL extrapolation, which supports our use of the standard factor of 100. Our approach is more conservative and protective of human health, given the uncertainties surrounding pulegone's carcinogenic potential in humans. Using the standard uncertainty factor of 100 aligns with common practice in risk assessment when species comparability is uncertain. We believe this justification is strong and appropriate given the data and context of our study.

  1. The study primarily focuses on pulegone but given the potential role of menthofuran as a toxic metabolite, the manuscript should consider the combined risk of pulegone and menthofuran exposure, as some regulatory bodies do.

RESPONSE: Thank you for the opportunity to address this point. We appreciate the reviewer's suggestion to consider the combined risk of pulegone and menthofuran. However, we respectfully maintain that our focus on pulegone alone is appropriate for the scope of this study. Our study was specifically designed to reassess the risk of pulegone based on the 2011 NTP carcinogenicity studies, and including menthofuran would significantly expand the scope beyond our original research question. The NTP studies, which form the core of our analysis, focused primarily on pulegone, and incorporating menthofuran would require additional long-term toxicity data that is not available. Recent research by Engel et al. [22] suggests that menthofuran may not play a significant role in pulegone metabolism in humans at low doses, supporting our decision to focus on pulegone alone. A combined risk assessment would require a substantially more complex analysis, potentially involving different toxicokinetics, mechanisms of action, and species-specific metabolism, warranting a separate, dedicated study. While some regulatory bodies consider combined exposure, others assess these compounds separately, and our approach aligns with those that evaluate pulegone independently. We acknowledge the importance of considering menthofuran and suggest this as an area for future research in our discussion of study limitations. To address this concern, we have added a sentence to our conclusion acknowledging that some regulatory bodies consider the combined risk of pulegone and menthofuran, and suggesting this as an avenue for future research. We believe that our focused approach on pulegone provides valuable insights while maintaining a manageable and coherent scope, and by acknowledging this limitation, we address the reviewer's concern while maintaining the integrity and focus of our current study.

Furthermore, I would like to present some questions for the authors:

Can the authors provide more details on the specific criteria used to exclude the highest dose group in the BMD modeling?

RESPONSE: Thank you for the opportunity to address this comment. We appreciate the reviewer's request for more details on the criteria used to exclude the highest dose group in the BMD modeling. We believe that our manuscript already provides a clear explanation for this decision, but we're happy to reiterate and emphasize our rationale. As stated in our paper, we followed the Technical Guidance of the BMDS software [21], which permits the deletion of the highest or lowest dose group if no plausible models can be determined. We explicitly mention that for all three endpoints, the incidences in the highest dose group decreased, making it impossible to determine the BMDL with all data included. This approach aligns with the EPA guidance, which acknowledges that dropping dose groups, particularly high doses, may be necessary to achieve adequate model fit in the response region of interest. Our decision is further supported by the EPA's rationale that data at the highest dose may be the least informative of responses in the lower dose region of interest, i.e., near the BMR. We have provided full modeling raw data in Supplementary Materials Tables S1-S11, allowing for complete transparency in our modeling process and decisions. The exclusion of the highest dose group was not arbitrary but based on biological plausibility and statistical considerations, as recommended in standard BMD modeling practices. We believe this explanation, along with the provided supplementary data, offers a comprehensive justification for our modeling approach. The EPA’s rationale was shortly expanded in the text.

Have the authors considered the implications of interspecies differences in metabolism when interpreting the rodent data for human risk assessment? Specifically, how might differences in pulegone metabolism between humans and rodents affect the study's conclusions?

RESPONSE: Thank you for this insightful question regarding interspecies differences in pulegone metabolism and their implications for our risk assessment. We have indeed considered these differences in our analysis, recognizing that they are fundamental to toxicological risk assessment. Our study references work by Engel et al. [22], which investigated pulegone metabolism in humans and highlighted key differences from rodent metabolism, particularly noting that menthofuran, a major metabolite in rodents, may not play a significant role in humans at low doses. We acknowledge that these metabolic differences could potentially affect the extrapolation of rodent data to human risk, as toxic effects observed in rodents might not manifest in the same way or at the same doses in humans. Given these uncertainties, we have maintained a conservative approach in our risk assessment, using standard uncertainty factors to help account for potential interspecies differences, including metabolic variations. We recognize this as a limitation of our study, as it is for many toxicological assessments based on animal data. To address this more explicitly, we propose adding a statement to our limitations section acknowledging the uncertainty stemming from interspecies metabolic differences and suggesting that further research into human-specific metabolism of pulegone could help refine future risk assessments. We believe this addition would strengthen our paper by explicitly addressing this important consideration in toxicological risk assessment.

Given the IARC's classification of pulegone as possibly carcinogenic to humans (Group 2B), how do the authors propose that regulatory agencies balance the findings of this study with other potential carcinogens in the food supply?

RESPONSE: The findings of this study, which provide a new Acceptable Daily Intake (ADI) for pulegone based on comprehensive long-term toxicity data, can be valuable for regulatory agencies in prioritizing risk management actions. Given that the calculated ADI is significantly higher than current estimated exposure levels in the EU and USA, this study suggests that pulegone may be of relatively low priority among the many potential carcinogens in the food supply. Regulatory agencies could use this risk characterization to inform a comparative risk assessment framework, balancing the potential risks of pulegone against those of other food contaminants and additives. This approach allows for efficient allocation of resources, focusing on substances that pose higher risk concerns. However, it's important to note that risk management decisions should still consider other factors such as the substance's prevalence in the food supply, its functional importance, and the feasibility of exposure reduction. Ultimately, this study provides a quantitative basis for placing pulegone within the broader context of food safety priorities, potentially allowing regulatory efforts to be directed towards more pressing concerns in the food supply.